# Exploring the Potential of Potato Peels for Bioethanol Production through Various Pretreatment Strategies and an In-House-Produced Multi-Enzyme System

Sanjeev Kumar Soni [1,*] , Binny Sharma [1], Apurav Sharma [1], Bishakha Thakur [1] and Raman Soni [2]

1   Department of Microbiology, Panjab University, Chandigarh 160014, India;
    binnysharma1409@gmail.com (B.S.); akashapurav5@gmail.com (A.S.); bishakhathakur1998@gmail.com (B.T.)
2   Department of Biotechnology, D.A.V. College, Chandigarh 160011, India; ramansoni@davchd.ac.in
*   Correspondence: sonisk@pu.ac.in

**Abstract:** This study aimed to explore the viability of converting potato peel waste into bioethanol using a custom-produced multi-enzyme preparation. Various pretreatment approaches were employed on the potato peels, including thermal, chemical, and thermo-chemical methods. These methods involved boiling for 30 and 60 min, steaming at different pressures and durations, and applying different concentrations of chemicals, including $H_2SO_4$, $HNO_3$, $CH_3COOH$, $HCl$, $NaOH$, $Ca(OH)_2$, $KOH$, $NH_3$, and $H_2O_2$, either individually or in combination with steam treatment. The pretreated potato peels were subsequently subjected to enzymatic hydrolysis using a crude multi-enzyme cocktail obtained from solid-state fermentation of wheat bran by a naturally occurring strain of *Aspergillus niger* P-19. This enzyme cocktail consisted of cellulases, hemicellulases, pectinase, and amylases. The most effective pretreatment combination involved the use of 3% $H_2SO_4$ followed by steam treatment under pressure, and enzymatic hydrolysis utilizing the crude multi-enzyme preparation. This combination resulted in the highest concentration of reducing sugars ($141.04 \pm 12.31$ g/L), with a carbohydrate conversion rate of 98.49% when a substrate loading of 20% was used. As a result, an ethanol yield of $43.2 \pm 3.82$ g/L, representing 21.6% based on dry matter, was achieved. Furthermore, supplementing the medium with peptone, $(NH_4)(H_2PO_4)$, and $ZnSO_4$ at a concentration of 0.1% *w/v* each, along with solid loadings of 22% and 24%, respectively, achieved yield improvements of 51.67 g/L and 54.75 g/L. However, the maximum productivity of 23.4% was observed with a 22% loading, compared to a yield of 22.8% with a 24% solid loading, based on dry matter.

**Keywords:** potato peels; pretreatment; in-house produced; multi-enzyme system; enzymatic saccharification; bioethanol

## 1. Introduction

The development of sustainable energy policies is a global priority aimed at promoting cleaner and more efficient energy supplies to address the energy crisis and mitigate global warming and air pollution. To achieve a sustainable future economy, innovative approaches to developing alternative energy sources are essential [1]. The production of biofuels from renewable resources is a viable technology for meeting increasing energy demands while reducing greenhouse gas emissions [2]. Agricultural biomass, primarily consisting of cellulose, hemicelluloses, and lignin, represents the most abundant renewable energy resource available today [3,4]. Large quantities of fruit and vegetable peels are generated in industries, households, and commercial areas, often mixed with other waste, rendering them unsuitable for further use. However, these waste peels can be effectively utilized for sugar recovery to produce biofuels.

Potatoes, scientifically known as *Solanum tuberosum*, are a widely consumed staple food globally, either in their natural form or processed into various products such as chips,

fries, mashed potatoes, and dehydrated products. However, during processing, a significant portion of the raw potato, ranging from 20% to 50%, is discarded as waste, primarily in the form of peels [5–8]. Potato peels consist of approximately 55% cellulose, 12% hemicellulose, and 14% lignin [9], along with varying amounts of starch and pectin. Typically, this food waste is disposed of in landfills, leading to environmental concerns such as groundwater pollution, unpleasant odors, and significant greenhouse gas emissions [10]. Therefore, it is crucial to manage potato peel waste in an eco-friendly manner, making it a significant concern for the potato industry [11]. Converting this waste into value-added products can address the issue of waste disposal and provide a promising solution to enhance the biofuel sector [12]. In recent years, there has been increasing interest in utilizing potato waste, including potato starch residue, waste potato mash, and potato peels, as feedstocks for bioethanol production [4,6–8,11,13–17]. Generally, the conversion of native biomass into bioethanol involves four stages: pretreatment, enzyme-catalyzed hydrolysis to generate fermentable sugars, fermentation to produce bioethanol, and product recovery [18–21].

The pretreatment stage plays a crucial role in ethanol production as it significantly impacts the subsequent stages. Pretreatment is necessary to disrupt the steric hindrance of lignin and hemicellulose [22], which enhances enzyme accessibility and improves cellulose digestibility. Various pretreatment techniques have been employed for different biomass residues, including physical methods such as hydrothermal treatment, steam explosion, microwave or ultrasonic wave-assisted treatments, chemical methods such as acid- and alkali-assisted treatments, and biological methods such as enzyme-assisted treatments. Each pretreatment method has a specific effect on the cellulose, hemicellulose, and lignin fractions, and the level of success varies for each method.

For the hydrolysis of pretreated biomass, both acids and enzymes can be used, but enzymes are preferred as they do not produce inhibitory compounds that can interfere with subsequent fermentation [23]. Enzyme-mediated hydrolysis also offers other advantages, such as lower energy consumption, mild conditions, and no corrosion. However, the use of a suitable combination of multiple enzyme systems with high specificity and cost-effectiveness remains a challenge for commercial applications [8,24–26]. Since potato peels contain cellulose, hemicelluloses, starch, and pectin, an enzyme cocktail consisting of carbohydrases, including endo-1,4-β-D-glucanase, exo-1,4-β-D-glucanase, β-1,4-glucosidase, endo-β-1,4-xylanase, endo-β-1,4-β-D-mannanase, α-amylase, gluco amylase, and pectinase, can be utilized to saccharify the biomass into free sugars. However, such a wide range of enzyme preparation is either not commercially available or is prohibitively expensive. Therefore, a strategy that involves in-house production of the enzyme cocktail using suitable microbes capable of co-producing the necessary enzyme systems could be advantageous in reducing the cost of multiple enzyme systems.

An ideal and effective pretreatment technique can greatly enhance the hydrolysis of biomass without generating any toxic by-products, thereby reducing the cost of ethanol production. Therefore, it is essential to optimize the pretreatment and enzymatic hydrolysis steps, including in-house production of multiple enzyme systems, to master the ethanol production process. In this study, we aim to investigate the most suitable pretreatment and low-cost enzymatic hydrolysis strategies using potato peel waste. Our goal is to develop a cost-effective, eco-friendly, and high-yield technology for bioethanol production.

## 2. Materials and Methods

### 2.1. In-House Production of Multi-Enzyme System

The multi-enzyme preparation, consisting of cellulases, hemicellulases, amylases, and pectinase, was obtained from solid-state fermentation of potato peels using a natural strain of *Aspergillus niger* P-19, as previously reported [21,27].

### 2.2. Quantitative Analysis of Dried Potato Peels

Potato peels, which are the major component of kitchen waste residues, were collected from homes and hostels of Panjab University, Chandigarh. The potato peels were dried in a

hot air oven at 70 °C overnight before quantitative analysis. The samples were pulverized using a laboratory mixer grinder with a power of 500 watts until the particles reached a size capable of passing through a 20-mesh sieve. The composition of the potato peels was determined by analyzing free-reducing sugar [28], total carbohydrates [29], cellulose [30], starch [31], and hemicellulose in terms of XGM (xylan + galactan + mannan) [32] using standard methods, as previously performed in a recent study by our research group [21].

### 2.3. Standardization of Various Thermal, Chemical, and Thermo-Chemical Strategies for Efficient Pretreatment of Potato Peels

Various physical and chemical pretreatment strategies were investigated, including boiling for different durations, steam treatment under varying pressure conditions with varying residence times, and acid/alkali treatments performed individually or in combination with steam.

### 2.3.1. Thermal Pretreatments

To prepare the samples, 10 g of powdered potato peels were added to 25 mL of distilled water in 250 mL Erlenmeyer flasks, resulting in a 40% solid loading. The flasks were then subjected to steam treatment in an autoclave under various pressure conditions for different time durations. These conditions included 10 psi for 15 min, 15 psi for 5 min, 15 psi for 15 min, 15 psi for 30 min, 15 psi for 45 min, and 15 psi for 60 min, each in separate flasks. For the boiling treatment, 10 g of the substrate was added to 25 mL of distilled water and placed in a boiling water bath for 30 min and 60 min, again in separate flasks.

### 2.3.2. Chemical Pretreatments

Twenty-five milliliters of different chemicals including $H_2SO_4$, $HNO_3$, $CH_3COOH$, HCl, $H_2O_2$, NaOH, $Ca(OH)_2$, KOH, and $NH_3$ (1% *v/v* or *w/v*) were added to separate flasks, each of which contained 10 g of powdered potato peels. The flasks were left at room temperature overnight and then neutralized with 1N NaOH/HCl.

### 2.3.3. Thermo-Chemical Pretreatments

The effects of 1.0–4.0% (*v/v* or *w/v*) of $H_2SO_4$, $HNO_3$, $CH_3COOH$, HCl, $H_2O_2$, NaOH, $Ca(OH)_2$, KOH, and $NH_3$ were studied in combination with steam under pressure. 10 g of powdered potato peels was dispensed into 25 mL of varying concentrations of chemical solutions taken in separate 250 mL Erlenmeyer flasks. The flasks were then subjected to steaming in an autoclave at 15 psi for 15 min and allowed to cool. The neutralization was performed using 1N NaOH/HCl.

### 2.4. Enzymatic Hydrolysis of Pretreated Potato Peel Residues Using In-House Multi-Enzyme Production

In order to achieve the efficient hydrolysis of potato peels, it is crucial to select an appropriate enzyme preparation that contains multiple enzyme systems to effectively handle the heterogeneous composition of the feedstock. The pretreated potato peels underwent enzymatic hydrolysis using an in-house multi-enzyme preparation obtained from *A. niger* P-19. The hydrolysis process was conducted at 50 °C and 150 rpm for 72 h, employing enzyme-to-substrate ratios of 16 IU CMCase, 2 IU FPase, 6 IU β-glucosidase, 2157 IU xylanase, 16 IU mannanase, 13 IU pectinase, 2220 U α-amylase, and 32 IU glucoamylase/g of substrate. The total volume was adjusted to 50 mL using 0.1 M acetate buffer (pH 4.5), resulting in a 20% solid loading. The samples were withdrawn at regular intervals of 24 h, centrifuged at 10,000 rpm (4 °C) for 10 min, and the supernatant was analyzed for total reducing sugars and glucose using the dinitrosalicylic acid (DNSA) and glucose oxidase-peroxidase methods, respectively [28,33]. The steam-pretreated sample (15 psi for 15 min) was used as the control, with its total volume also adjusted to 50 mL using the buffer. The carbohydrate conversion was calculated as the percentage of theoretical reducing sugar yield, determined through the breakdown of polysaccharides into sugar, as defined by Equation (1).

$$(C_6H_{10}O_5)_n + {}_nH_2O \rightarrow (C_6H_{12}O_6)_n \tag{1}$$

Carbohydrate conversion (%) = [Reducing sugars]/(1.11× f × [Biomass]) × 100

Reducing sugars represent the concentration of total reducing sugars in grams, biomass represents the mass of dry pretreated potato peels, f represents the carbohydrate fraction (in terms of glucose) in dry biomass in grams per gram, and 1.11 is the factor that accounts for the mass balance in the conversion of polysaccharides to sugars. The reducing sugars and glucose yields have been expressed in terms of the mass of total reducing sugars and glucose produced per mass of pretreated potato peels, as well as per volume of the hydrolysate.

### 2.5. Qualitative Analysis of Free Sugars in the Enzymatic Hydrolysate of Potato Peels Using Thin-Layer Chromatography (TLC)

The sugars obtained following the enzymatic hydrolysis of potato peels, which were pretreated with a combination of 3% $H_2SO_4$ and steam, were subjected to analysis using TLC (thin-layer chromatography) according to the standard protocol [34]. For this analysis, 10 μL of suitably diluted samples of the standard sugars, including glucose, galactose, maltose, xylose, mannose, and arabinose, along with the hydrolysate of potato peels, were applied onto a silica gel plate (60 F254, Merck KGaA, Darmstadt, Germany) as bands using an automated TLC sampler. The plates were subsequently positioned inside a glass jar containing a developing reagent consisting of butanol, acetic acid, and water in a 3:1:1 ratio. The sugars were detected using a mixture of N-(1-Naphthyl)ethylenediamine (NED), methanol, and $H_2SO_4$, with specific quantities of 0.30 g, 95 mL, and 5 mL, respectively.

### 2.6. Structural Changes in Untreated, Thermo-Acidic Pretreated, and Enzymatically Hydrolyzed Samples of Potato Peels

The effectiveness of the steam pretreatment, thermo-acidic pretreatment, and enzymatic hydrolysis of potato peels was analyzed by examining untreated, thermo-acidic pretreated, and enzymatically hydrolyzed samples using scanning electron microscopy (SEM) and X-ray diffraction (XRD) techniques. The services of the Sophisticated Analytical Instrumentation Facility at the Central Instrumentation Laboratory (CIL), Panjab University, Chandigarh, were utilized for this purpose.

### 2.7. Fermentation of Sugars Released after Enzymatic Hydrolysis of Thermo-Chemically Pretreated Potato Peels

All flasks containing hydrolyzed mashes of pretreated biomass residues were inoculated with a yeast cell pellet derived from a 10% *v/v* suspension of the distiller's strain of *Saccharomyces cerevisiae* HT, already available in the laboratory. The yeast cells were obtained by centrifuging at 10,000 rpm for 10 min at 4 °C. Prior to inoculation, the yeast cells were cultured overnight in YPD broth at 28 °C with shaking at 200 rpm, resulting in a viable cell count of $1.0 \times 10^8$/mL. The inoculated flasks were then incubated under stationary conditions in a BOD incubator at 30 °C for 72 h. The alcohol content was determined using the potassium dichromate method [35]. The fermentation efficiency was expressed as a percentage of the theoretical conversion of hexoses to ethanol, calculated based on the sugars actually utilized and the actual ethanol obtained, as indicated by Equation (2).

$$C_6H_{12}O_6 \rightarrow 2C_2H_5OH + 2CO_2 \tag{2}$$

The ethanol obtainable from the sugars, as per Equation (2), is 51.1 g/100 g of glucose.

Ethanol yields were reported in two units: mass of alcohol produced per volume of potato peel hydrolysate and mass of alcohol produced per mass of dry potato peels. The fermentation efficiency was calculated using the following formula:

Fermentation Efficiency = (Actual Ethanol Yield/Theoretical Ethanol Yield) × 100

The actual ethanol yield is the mass of ethanol produced during fermentation, while the theoretical ethanol yield represents the maximum possible yield based on the amount of sugars used in the fermentation process. This calculation allows us to determine the efficiency of the fermentation process in converting sugars into ethanol and is expressed as a percentage of the theoretical maximum yield.

### 2.8. Effect of the Supplementation of Nutrients in the Hydrolysate on Fermentation and Substrate Loading

Various nutrients, including $MgSO_4.7H_2O$, $(NH_4)_2SO_4$, yeast extract, urea, peptone, $(NH_4)_2(HPO_4)$, $(NH_4)(H_2PO_4)$, $KH_2PO_4$, $K_2HPO_4$, and $ZnSO_4$, have been employed in attempts to enhance alcohol productivity. To carry out this study, 100 g of powdered potato peels were placed in a 1000 mL Erlenmeyer flask with 250 mL of 3% $H_2SO_4$. The mixture was subjected to steaming in an autoclave at 15 psi for 15 min and then allowed to cool. The pH was then neutralized using 1N NaOH, followed by the addition of an enzyme preparation to achieve the enzyme: substrate ratio described in Section 2.4. The final volume was adjusted to 500 mL with 0.1 M acetate buffer, pH 4.5, to maintain a 20% solid loading. The flask was incubated at 50 °C in a water bath shaker at 150 rpm for 72 h, and the resulting sample was centrifuged at 10,000 rpm for 10 min. The supernatant was analyzed for total reducing sugars and glucose released using the dinitrosalicylic acid (DNSA) and glucose oxidase-peroxidase methods, respectively [28,33].

The potato peel hydrolysate was distributed equally into separate 150 mL Erlenmeyer flasks and inoculated, separately, with a pellet of yeast cells obtained after centrifugation of a 10% *v/v* cell suspension of a distiller's strain of *S. cerevisiae* HT. The yeast cells were cultured overnight in YPD broth at 28 °C with shaking at 200 rpm, resulting in a viable cell count of $1.0 \times 10^8$/mL. Each flask was supplemented individually with the nutrient sources mentioned above. The flasks were then incubated under stationary conditions in a BOD incubator at 30 °C for 72 h, and the alcohol content was determined using the potassium dichromate method [35]. Additionally, the effect of the solid concentration was studied by fermenting the hydrolysates supplemented with peptone + $(NH_4)(H_2PO_4)$ + $ZnSO_4$ (0.1% *w/v* each) with substrate loadings of 22% and 24%. The overall efficiency of the process was determined on the basis of the carbohydrate and fermentation efficiencies as per the following formula:

$$\text{Overall efficiency}(\%) = \frac{\text{Carbohydrate conversion efficiency} \times \text{Fermentation efficiency}}{100}$$

## 3. Results

To achieve efficient and cost-effective degradation of agricultural and agro-food waste residues, pretreatment plays a crucial role in determining the final product yield. Recent advancements in pretreatment strategies have revolutionized the biofuel industry by increasing the porosity of agricultural and agro-food waste residues, thereby reducing cellulose crystallinity. This enhancement in porosity facilitates enzymatic attack and promotes the release of sugars. In this study, we evaluated various pretreatment methods followed by enzymatic hydrolysis using an effective enzyme cocktail consisting of multiple hydrolytic carbohydrases and fermentation of released sugars with *S. cevevisiae*. The aim was to enhance the yield of fermentable sugars and bioethanol from potato peel waste.

### 3.1. In-House Production of Multi-Enzyme System

The solid-state fermentation of potato peels resulted in a multi-enzyme preparation, which was obtained by extracting the moldy peels with distilled water. The enzyme preparation exhibited individual activities of 10.38, 1.44, 4.06, 104.95, 10.41, 8.59, and 21.21 IU/mL for CMCase, FPase, β-glucosidase, xylanase, mannanase, pectinase, and glucoamylase, respectively. Additionally, it displayed an α-amylase activity of 1480 U/mL. The optimal temperature and pH for this enzyme cocktail were determined to be 50 °C and pH 4.5, respectively, as previously described in studies [21,36].

### 3.2. Composition Analysis of Dried Potato Peels

The total carbohydrate content of the potato peels was estimated to be 71.6% on a dry weight basis, consisting of 7.47% free sugars, 20.2% cellulose, 19.4% starch, and 15.2% hemicelluloses. This high carbohydrate content makes potato peels an attractive substrate for ethanol production.

### 3.3. Evaluation and Standardization of Pretreatments of Dried Potato Peels

The primary goal of pretreatment is to eliminate lignin and hemicellulose while decreasing the crystallinity of cellulose, which makes it more accessible for enzymatic degradation. However, the suitability of the pretreatment process is highly dependent on the feedstock used.

#### 3.3.1. Thermal Pretreatment

Thermal pretreatment methods encompass a range of techniques, including conventional heating, hydrothermal treatment, steam explosion, and microwave irradiation, typically performed within a temperature range of 50 to 240 °C [37]. However, one significant drawback of thermal pretreatment is the generation of soluble phenolic compounds when the biomass is exposed to temperatures exceeding 160 °C [38]. To mitigate this issue, some researchers have employed a moderate temperature of 120 °C for the thermal pretreatment of sawdust [39] and safflower straw [40], with residence times of 15 and 60 min, respectively. In the present study, several thermal techniques were employed, including boiling, steam treatment under pressure, and microwave irradiation, with varying residence times. While boiling the substrate for 30 and 60 min showed improved outcomes compared to the untreated substrate, steam treatment at 15 psi for 15 min yielded higher amounts of reducing sugars (359.43 ± 32.25 mg/gds) and glucose (288.45 ± 21.64 mg/gds), with a carbohydrate conversion efficiency of 50%. Consequently, this method was deemed the most effective among all the thermal strategies adopted (Table 1). This finding aligns with a previous study conducted by our research group, where maximum sugar release was achieved from de-oiled rice bran using steam pretreatment at 15 psi for 15 to 60 min [21]. Steam pretreatment has the potential to disrupt the lignin and hemicellulose components of the biomass's crystalline structure, rendering cellulose more susceptible to enzymatic digestion. These results are consistent with a study where the enzymatic conversion of steam-pretreated corn stover increased fourfold compared to untreated material [41]. The impact of longer exposure times under steam pressure was also examined, but no significant changes were observed in the conversion efficiency (Table 1). Considering the insignificant difference in sugar release, extending the pretreatment period from 15 to 60 min was deemed inefficient, leading to the selection of 15 min as the optimal pretreatment duration for subsequent research.

**Table 1.** Total reducing sugar yields as a result of thermal pretreatment of potato peels.

| Thermal Treatment | Total Reducing Sugars Including Glucose Yields at Different Time Intervals (mg/g) | | | | Total Concentration (g/L) | Carbohydrate Conversion Efficiency (%) |
|---|---|---|---|---|---|---|
| | 0 h | 24 h | 48 h | 72 h | | |
| Boiling—30 min | 52.30 ± 4.92 (40.03 ± 3.94) | 186.15 ± 15.43 * (125.00 ± 10.14) * | 198.76 ± 18.45 * (150.00 ± 11.41) | 218.76 ± 17.45 * (184.33 ± 11.41) | 43.752 ± 3.78 * (36.87 ± 3.21) | 30.55 ± 2.43 |
| Boiling—60 min | 61.53 ± 5.78 (49.75 ± 4.01) | 200.00 ± 17.16 * (133.33 ± 10.12) * | 238.46 ± 19.54 (162.33 ± 12.32) | 278.46 ± 26.54 (190.67 ± 16.14) | 55.692 ± 3.58 (38.13 ± 3.02) | 38.89 ± 3.70 |
| Steam at 10 psi—15 min | 60.00 ± 4.89 (48.21 ± 4.12) | 203.84 ± 17.63 * (150.23 ± 11.23) | 286.15 ± 18.52 (198.68 ± 17.16) | 306.15 ± 25.12 (210.53 ± 18.71) | 61.23 ± 4.07 (42.11 ± 3.94) | 42.76 ± 3.50 |
| Steam at 15 psi—5 min | 57.67 ± 3.25 (39.62 ± 3.02) | 210.00 ± 17.65 * (138.75 ± 11.44) * | 215.38 ± 17.96 * (150.77 ± 12.32) * | 295.88 ± 24.74 (193.75 ± 17.23) | 59.176 ± 4.65 (38.75 ± 3.64) | 41.32 ± 3.45 |

**Table 1.** *Cont.*

| Thermal Treatment | Total Reducing Sugars Including Glucose Yields at Different Time Intervals (mg/g) | | | | Total Concentration (g/L) | Carbohydrate Conversion Efficiency (%) |
|---|---|---|---|---|---|---|
| | 0 h | 24 h | 48 h | 72 h | | |
| Steam at 15 psi—15 min | 77.69 ± 5.24 (65.76 ± 6.13) | 270.85 ± 21.54 (199.03 ± 16.41) | 316.92 ± 26.18 (228.28 ± 20.21) | 359.43 ± 32.25 (288.45 ± 21.64) | 71.886 ± 5.97 (57.69 ± 4.15) | 50.20 ± 4.50 |
| Steam at 15 psi—30 min | 83.39 ± 7.54 (67.40 ± 6.02) | 275.433 ± 21.76 (203.53 ± 17.14) | 319.51 ± 25.43 (228.43 ± 21.24) | 360.85 ± 31.25 (294.51 ± 21.65) | 72.17 ± 6.12 (58.90 ± 4.62) | 50.40 ± 4.36 |
| Steam 15 psi—45 min | 85.19 ± 7.14 (67.71 ± 6.17) | 283.30 ± 24.78 (218.29 ± 20.23) | 320.29 ± 25.76 (235.42 ± 20.41) | 360.96 ± 35.23 (297.00 ± 22.31) | 72.192 ± 5.96 (59.40 ± 4.71) | 50.41 ± 4.92 |
| Steam 15 psi—60 min | 87.48 ± 6.23 (77.40 ± 6.65) | 296.92 ± 21.24 (249.02 ± 21.64) | 320.51 ± 26.12 (260.55 ± 21.41) | 362.04 ± 30.25 (297.51 ± 22.54) | 72.408 ± 6.24 (59.50 ± 4.84) | 50.56 ± 4.22 |
| Untreated (control) | 17.66 ± 1.09 (10.92 ± 9.14) | 173.24 ± 14.38 (116.12 ± 10.14) | 176.03 ± 16.37 (120.25 ± 11.41) | 210.20 ± 19.50 (123.65 ± 10.21) | 42.04 ± 3.21 (24.73 ± 2.14) | 29.36 ± 2.72 |

The values in parentheses indicate the glucose yields. All the values differ from the control significantly by the Holm–Sidak test with $p < 0.001$, except those marked with *.

### 3.3.2. Chemical Pretreatments

The results in Table 2 indicate that steam under pressure at 15 psi for 15 min was superior to chemical treatment in terms of conversion efficiency, as no chemical treatment was able to match the performance of the former. Therefore, it can be concluded that steam under pressure is a necessary requirement for the pretreatment of dried potato peels.

**Table 2.** Total reducing sugars and glucose yields as a result of chemical pretreatments of potato peels followed by enzymatic hydrolysis.

| Chemical Treatment | Reducing Sugars Including Glucose Yields at Different Time Intervals (mg/g) | | | | Total Concentration (g/L) | Carbohydrate Conversion Efficiency (%) |
|---|---|---|---|---|---|---|
| | 0 h | 24 h | 48 h | 72 h | | |
| $H_2SO_4$ | 70.28 ± 6.63 (55.15 ± 5.01) | 150.21 ± 13.24 (126.21 ± 10.42) | 167.65 ± 12.63 (132.55 ± 11.41) | 177.35 ± 15.14 (137.45 ± 11.41) | 35.47 ± 2.98 (27.49 ± 2.01) | 24.76 ± 2.11 |
| $HNO_3$ | 63.61 ± 5.36 (49.21 ± 4.56) | 143.05 ± 13.23 (120.15 ± 11.21) | 150.12 ± 11.25 (127.21 ± 11.32) | 150.38 ± 12.32 (130.12 ± 12.31) | 30.07 ± 2.45 (26.02 ± 2.04) | 21.00 ± 1.72 |
| $CH_3COOH$ | 57.04 ± 4.75 (40.12 ± 3.64) | 132.85 ± 11.24 (112.35 ± 10.45) | 134.57 ± 13.14 (116.27 ± 10.47) | 135.03 ± 10.89 (117.0 ± 10.62) | 27.00 ± 2.14 (23.4 ± 2.12) | 18.85 ± 1.52 |
| HCl | 60.97 ± 4.03 (43.57 ± 3.42) | 135.54 ± 10.56 (117.51 ± 10.64) | 139.27 ± 12.25 (119.17 ± 10.95) | 140.22 ± 12.05 (120.35 ± 10.14) | 28.04 ± 1.98 (24.07 ± 2.04) | 19.58 ± 1.68 |
| $H_2O_2$ | 41.01 ± 3.86 (33.01 ± 3.02) | 100.95 ± 9.86 (88.05 ± 8.10) | 105.04 ± 8.23 (90.04 ± 8.74) | 106.93 ± 10.03 (91.93 ± 8.75) | 21.38 ± 1.98 (18.38 ± 1.56) | 14.93 ± 1.40 |
| NaOH | 67.47 ± 4.98 (50.12 ± 4.97) | 145.50 ± 12.96 (120.91 ± 10.14) | 158.36 ± 14.58 (130.16 ± 11.23) | 159.12 ± 14.05 (130.22 ± 11.36) | 31.82 ± 2.87 (26.04 ± 2.12) | 22.22 ± 1.96 |
| $Ca(OH)_2$ | 54.24 ± 5.10 (40.24 ± 3.87) | 130.03 ± 12.45 (110.12 ± 10.32) | 135.14 ± 12.03 (115.24 ± 10.08) | 137.15 ± 11.23 (117.25 ± 11.01) | 27.43 ± 2.14 (23.45 ± 2.26) | 19.15 ± 1.56 |
| KOH | 51.13 ± 3.96 (38.13 ± 3.64) | 135.50 ± 13.12 (115.50 ± 10.02) | 136.28 ± 11.06 (116.28 ± 10.41) | 142.05 ± 11.87 (119.01 ± 11.02) | 28.41 ± 2.01 (23.80 ± 2.14) | 19.83 ± 1.65 |
| $NH_3$ | 50.12 ± 4.54 (38.10 ± 3.54) | 133.29 ± 11.49 (116.15 ± 10.14) | 140.45 ± 13.47 (117.25 ± 10.21) | 145.35 ± 13.24 (118.25 ± 10.64) | 29.07 ± 2.54 (23.65 ± 2.16) | 20.30 ± 1.84 |
| Steam at 15 psi—15 min (control) | 150.11 ± 12.21 (129.11 ± 10.21) | 254.26 ± 21.23 (204.16 ± 17.41) | 334.65 ± 29.85 (304.15 ± 26.21) | 342.43 ± 29.23 (307.23 ± 27.98) | 68.48 ± 5.45 (41.44 ± 3.01) | 47.82 ± 4.08 |

The values in parentheses indicate the glucose yields. All the values differ from the control significantly by the Holm–Sidak test.

### 3.3.3. Thermo-Chemical Pretreatments

Numerous studies have demonstrated the effectiveness of thermo-chemical treatments using dilute acids or bases for pretreating agricultural and agro-food waste residues. Among these treatments, $H_2SO_4$, known for its cost-effectiveness and efficacy, is frequently used at concentrations below 4%. However, when acids or bases were used alone at a 1% concentration, they did not yield significant improvements in enzymatic hydrolysis compared to steam treatment. To investigate the impact of different concentrations of acids

and bases, varying concentrations (ranging from 1% to 4%) of all acids and bases were combined with steam for the pretreatment of dried potato peels as depicted in Table 3. The combination of chemical agents and steam exhibited enhanced total reducing sugar release, with thermo-acidic treatments proving more effective than thermo-alkali treatments. Notably, $H_2SO_4$ resulted in the highest sugar release, particularly glucose. At a concentration of 1%, $H_2SO_4$ released $525.00 \pm 39.65$ mg/g of reducing sugars, including $399.00 \pm 31.23$ mg/g of glucose, after 72 h of enzymatic hydrolysis. As the concentration of $H_2SO_4$ was increased to 2% and 3%, the sugar yield further improved, reaching $635 \pm 59.47$ mg/g and $705.20 \pm 66.36$ mg/g, respectively. This resulted in enhanced conversion efficiencies of 88.68% and 98.49%, accompanied by sugar concentrations of $127.00 \pm 11.14$ g/L and $141.04 \pm 12.31$ g/L, respectively (Table 4). A similar trend was observed with other acids, although $H_2SO_4$ demonstrated the highest effectiveness, as indicated in Tables 3 and 4.

**Table 3.** Total reducing sugars and glucose yields as a result of thermo-chemical pretreatment of potato peels with varying concentrations of chemicals followed by enzymatic hydrolysis.

| Treatments | Chemical | Conc. | Total Reducing Sugars Including Glucose Yields at Different Time Intervals (mg/g) | | | | Carbohydrate Conversion Efficiency (%) |
|---|---|---|---|---|---|---|---|
| | | | 0 h | 24 h | 48 h | 72 h | |
| Chemical + steam at 15 psi—15 min + enzymatic treatment | $H_2SO_4$ | 1% | 124.38 ± 11.32 (60 ± 5.64) * | 382.24 ± 26.54 * (324 ± 30.02) | 390.65 ± 36.21 * (328 ± 31.31) | 525.00 ± 39.65 * (399.00 ± 31.23) * | 73.32 ± 5.53 * |
| | HNO₃ | | 105.61 ± 9.23 * (59.25 ± 5.12) * | 328.08 ± 20.25 * (202.59 ± 19.47) * | 330.12 ± 25.23 (254.81 ± 20.41) * | 500.40 ± 35.23 * (302.22 ± 16.52) * | 69.88 ± 4.92 * |
| | CH₃COOH | | 139.04 ± 12.45 (81.48 ± 7.45) | 325.84 ± 21.54 * (218.89 ± 15.03) * | 327.87 ± 20.25 (285.18 ± 20.01) * | 355.33 ± 22.15 (325.92 ± 24.02) * | 49.62 ± 3.09 |
| | HCl | | 102.97 ± 9.45 * (62.67 ± 6.01) * | 369.53 ± 22.38 * (206.37 ± 17.97) * | 377.77 ± 20.23 * (222.81 ± 15.24) * | 380.00 ± 22.24 * (327.25 ± 20.25) * | 53.07 ± 3.11 * |
| | $H_2O_2$ | | 74.71 ± 6.54 * (31.67 ± 2.74) * | 310.95 ± 20.23 * (240.80 ± 16.42) * | 355.04 ± 20.41 * (309.25 ± 21.34) * | 386.93 ± 25.25 * (310.35 ± 20.24) * | 54.04 ± 3.52 * |
| | NaOH | | 94.43 ± 8.75 * (60.74 ± 5.64) * | 345.50 ± 20.25 * (203.77 ± 13.12) * | 349.46 ± 22.14 (261.18 ± 17.40) * | 368.53 ± 21.56 (328 ± 20.01) * | 51.47 ± 3.01 |
| | Ca(OH)₂ | | 94.21 ± 7.85 * (63.70 ± 6.01) * | 343.03 ± 26.54 * (235.67 ± 20.34) * | 355.14 ± 32.75 * (286.70 ± 21.24) * | 380.35 ± 30.29 * (331.85 ± 31.29) * | 53.12 ± 4.24 * |
| | KOH | | 92.13 ± 8.54 * (60.74 ± 5.71) * | 345.50 ± 31.25 * (230.51 ± 20.12) * | 360.78 ± 32.14 * (288.81 ± 26.13) * | 362.05 ± 31.25 (328 ± 30.24) * | 50.56 ± 4.38 |
| | NH₃ | | 101.12 ± 9.21 * (62.22 ± 5.06) * | 323.59 ± 28.65 * (210.59 ± 20.01) * | 343.45 ± 30.21 (213.33 ± 20.07) * | 359.55 ± 32.54 (331.25 ± 31.69) * | 50.21 ± 4.56 |
| | $H_2SO_4$ | 2% | 465.67 ± 35.45 (240.37 ± 2.13) | 590.33 ± 50.21 (411.11 ± 40.14) | 605.33 ± 52.75 (426.67 ± 40.01) | 635 ± 59.47 (440.45 ± 31.62) | 88.68 ± 8.32 |
| | HNO₃ | | 438.4 ± 32.54 (304 ± 27.14) | 599.06 ± 52.31 (413.62 ± 22.31) | 604 ± 52.14 (420.44 ± 20.13) | 633.33 ± 61.04 (429.15 ± 24.32) | 88.45 ± 8.52 |
| | CH₃COOH | | 248.27 ± 21.24 (137.92 ± 11.04) | 490.06 ± 41.02 (324.67 ± 31.41) | 530 ± 51.21 * (355.77 ± 32.19) | 571.65 ± 56.24 * (464.55 ± 41.26) | 79.83 ± 7.87 * |
| | HCl | | 320 ± 24.54 (188.44 ± 17.13) | 418.81 ± 36.21 * (380.44 ± 32.32) | 428 ± 40.27 * (384 ± 30.23) | 496.00 ± 45.12 * (398.22 ± 31.65) * | 69.27 ± 6.31 * |
| | $H_2O_2$ | | 166.73 ± 14.23 (99.40 ± 8.23) | 407.60 ± 34.21 * (115.11 ± 10.23) | 549.00 ± 50.02 * (269.92 ± 12.13) * | 569.33 ± 50.23 * (371.63 ± 20.24) * | 79.51 ± 7.03 * |
| | NaOH | | 196.67 ± 17.44 (114.81 ± 10.21) | 403.33 ± 36.21 * (362.96 ± 31.04) | 450 ± 41.78 * (348.11 ± 34.01) | 550.00 ± 54.65 * (407.40 ± 36.46) * | 76.81 ± 7.65 * |
| | Ca(OH)₂ | | 143.07 ± 11.34 (82.22 ± 8.01) | 400.66 ± 37.87 * (328.89 ± 30.02) | 401 ± 34.56 * (345.33 ± 31.98) | 456.33 ± 41.24 * (389.15 ± 36.12) * | 63.73 ± 5.77 * |
| | KOH | | 185.61 ± 15.52 (110.22 ± 10.02) | 404.43 ± 35.23 * (337.33 ± 31.32) | 448 ± 40.31 * (355.56 ± 32.35) | 464.00 ± 42.36 * (398.22 ± 35.23) * | 64.80 ± 5.93 * |
| | NH₃ | | 186.67 ± 16.02 (124.44 ± 11.24) | 309.33 ± 26.14 * (303.62 ± 26.13) | 390.67 ± 30.23 * (399.03 ± 36.12) | 448.00 ± 41.54 * (400.00 ± 35.23) * | 62.56 ± 5.82 * |

**Table 3.** *Cont.*

| Treatments | Chemical | Conc. | Total Reducing Sugars Including Glucose Yields at Different Time Intervals (mg/g) | | | | Carbohydrate Conversion Efficiency (%) |
| --- | --- | --- | --- | --- | --- | --- | --- |
| | | | 0 h | 24 h | 48 h | 72 h | |
| | $H_2SO_4$ | | 587.8 ± 48.23 (407.02 ± 38.23) | 610.4 ± 52.13 (407.02 ± 38.24) | 658.4 ± 60.12 (424.04 ± 36.12) | 705.20 ± 66.36 (475.60 ± 31.02) | 98.49 ± 9.29 |
| | $HNO_3$ | | 527.4 ± 40.03 (358.08 ± 32.13) | 532.4 ± 48.52 (358.08 ± 30.24) | 564.4 ± 51.78 * (434.46 ± 41.02) | 666.40 ± 58.97 (441.25 ± 41.23) | 93.07 ± 8.26 |
| | $CH_3COOH$ | 3% | 442.86 ± 41.36 (310.63 ± 30.02) | 485.73 ± 40.36 (310.63 ± 30.03) | 651.06 ± 63.21 (427.37 ± 41.04) | 681.33 ± 64.24 (476.31 ± 46.23) | 95.15 ± 8.99 |
| | HCl | | 440 ± 40.24 (245.74 ± 21.30) | 540.67 ± 46.98 (245.74 ± 21.24) * | 594 ± 57.12 (382.26 ± 35.62) | 608.65 ± 55.24 * (390.05 ± 36.23) * | 85.00 ± 7.73 * |
| | $H_2O_2$ | | 310 ± 26.41 (205.67 ± 17.21) | 413.33 ± 40.24 * (205.67 ± 13.35) * | 516.66 ± 48.79 * (300.60 ± 29.15) * | 578.30 ± 51.02 * (302.40 ± 50.01) * | 80.76 ± 7.14 * |
| | NaOH | | 296 ± 15.23 (132.05 ± 11.02) | 365.87 ± 21.03 * (132.05 ± 11.24) * | 487.46 ± 24.78 (221.56 ± 21.24) * | 553.20 ± 48.94 * (250.30 ± 26.23) | 77.26 ± 6.85 * |
| | $Ca(OH)_2$ | | 272.8 ± 24.26 (165.39 ± 14.21) | 300.53 ± 26.41 * (165.39 ± 13.45) * | 305.06 ± 26.13 (260.92 ± 24.14) * | 472.65 ± 41.05 * (280.85 ± 21.20) * | 66.01 ± 5.73 * |
| | KOH | | 221 ± 21.01 (138.29 ± 12.31) | 376.67 ± 30.24 * (138.29 ± 12.45) * | 498.66 ± 32.78 * (340.25 ± 31.69) * | 554.30 ± 42.53 * (350.35 ± 27.12) * | 77.41 ± 5.95 * |
| | $NH_3$ | | 297.6 ± 20.24 (204.25 ± 18.21) | 401.6 ± 36.13 * (204.25 ± 18.12) * | 499.2 ± 41.21 * (233.61 ± 20.08) * | 582.40 ± 48.91 * (245.05 ± 22.13) * | 81.34 ± 6.85 * |
| | $H_2SO_4$ | | 450 ± 41.02 (166.67 ± 12.31) | 646.67 ± 61.24 (361.70 ± 35.23) | 663.33 ± 61.02 (411.34 ± 34.23) | 666.50 ± 61.21 (457.15 ± 34.22) | 93.08 ± 8.57 |
| | $HNO_3$ | | 516.67 ± 48.20 (287.26 ± 24.32) | 586.67 ± 51.42 (343.97 ± 31.23) | 646.67 ± 60.24 (375.88 ± 28.12) | 654.00 ± 60.24 (430.70 ± 31.36) | 91.34 ± 8.43 |
| | $CH_3COOH$ | 4% | 313.6 ± 30.23 (170.28 ± 15.94) | 490 ± 41.63 (319.71 ± 20.02) | 601.07 ± 56.54 (396.17 ± 24.36) | 620.65 ± 55.24 * (462.00 ± 31.23) | 86.68 ± 7.73 * |
| | HCl | | 330 ± 31.42 (95.74 ± 8.92) | 510 ± 47.24 (322.38 ± 30.68) | 540 ± 50.02 * (380.36 ± 29.56) | 582.00 ± 51.84 * (385.72 ± 26.23) * | 81.28 ± 7.26 * |
| | $H_2O_2$ | | 388.27 ± 36.01 (140.70 ± 1.32) | 486.4 ± 45.75 (222.34 ± 21.32) * | 522.93 ± 50.63 * (310.21 ± 21.62) * | 597.33 ± 53.43 * (362.00 ± 30.04) * | 83.42 ± 7.48 * |
| | NaOH | | 221 ± 20.28 (108.51 ± 9.24) | 299.2 ± 25.54 * (109.7 ± 9.13) | 406 ± 28.54 * (180.8 ± 16.21) | 510.00 ± 47.12 * (262.25 ± 24.41) * | 71.22 ± 6.60 * |
| | $Ca(OH)_2$ | | 282 ± 21.04 (123.33 ± 11.02) | 313.33 ± 27.87 * (180 ± 10.45) * | 319.6 ± 29.34 (200 ± 19.01) * | 438.67 ± 40.23 * (201.43 ± 19.02) | 61.26 ± 5.63 * |
| | KOH | | 194.27 ± 15.64 (163.33 ± 15.65) | 419.87 ± 40.12 * (180 ± 10.32) * | 502.67 ± 45.24 * (360 ± 34.23) | 526.40 ± 51.45 * (376.00 ± 32.21) * | 73.51 ± 7.18 * |
| | $NH_3$ | | 288 ± 25.65 (187.23 ± 1.64) | 512 ± 50.45 (300.42 ± 26.45) | 531.2 ± 51.76 * (326.80 ± 30.23) | 588.80 ± 52.76 * (342.86 ± 31.34) * | 82.23 ± 7.36 * |
| Steam + enzyme treatment | | | 59.21 ± 5.23 (32.15 ± 2.64) | 364.26 ± 30.26 (207.03 ± 19.87) | 464.33 ± 32.45 (251.11 ± 21.24) | 503.20 ± 31.54 (336.00 ± 24.21) | 70.27 ± 4.40 |

The values in parentheses indicate the glucose yields. All the values differ from the control significantly by the Holm–Sidak test with *p* < 0.001, except those marked with *.

**Table 4.** Ethanol yields as a result of thermo-chemical pretreatment of potato peels with varying concentrations of chemicals followed by enzymatic hydrolysis.

| Treatments | Chemical | Conc. | Initial Sugars (g/L) | Residual Sugars (g/L) | Ethanol (g/L) | Fermentation Efficiency (%) | Ethanol Yield (mg/g) | Overall Efficiency (%) |
| --- | --- | --- | --- | --- | --- | --- | --- | --- |
| Chemical + steam at 15 psi—15 min + enzymatic treatment | $H_2SO_4$ | | 105.91 ± 7.45 * (68.8 ± 6.32) * | 25.2 ± 1.89 (0.0) | 26.4 ± 1.82 * | 64.01 ± 0.05 | 132 ± 9.1 * | 46.94 ± 0.002 |
| | $HNO_3$ | | 100.08 ± 6.87 * (60.44 ± 5.24) * | 30.64 ± 2.89 * (0.0) | 20.5 ± 1.18 * | 57.77 ± 0.25 | 102.5 ± 5.9 * | 40.38 ± 0.012 |
| | $CH_3COOH$ | 1% | 71.06 ± 6.03 (65.18 ± 6.12) * | 4.88 ± 0.35 (0.0) | 10.75 ± 0.92 | 31.79 ± 0.09 | 53.75 ± 4.60 | 15.78 ± 0.002 |
| | HCl | | 77.38 ± 7.14 * (62.07 ± 6.13) * | 9.55 ± 0.93 (0.0) | 13.7 ± 1.25 | 39.53 ± 0.14 | 68.5 ± 6.25 | 20.98 ± 0.004 |
| | $H_2O_2$ | | 77.34 ± 6.98 * (62.07 ± 6.07) * | 13.31 ± 0.93 (0.0) | 19.5 ± 1.84 * | 59.60 ± 0.08 | 97.5 ± 9.20 * | 32.21 ± 0.002 |
| | NaOH | | 73.70 ± 7.02 (65.6 ± 6.03) * | 7.1 ± 0.78 (0.0) | 19.9 ± 1.86 * | 58.47 ± 0.14 | 99.5 ± 9.30 * | 30.10 ± 0.004 |

Table 4. *Cont.*

| Treatments | Chemical | Conc. | Initial Sugars (g/L) | Residual Sugars (g/L) | Ethanol (g/L) | Fermentation Efficiency (%) | Ethanol Yield (mg/g) | Overall Efficiency (%) |
|---|---|---|---|---|---|---|---|---|
| | $Ca(OH)_2$ | | 76.07 ± 6.94 (66.37 ± 6.12) * | 8.7 ± 0.76 (0.0) | 19.6 ± 1.78 * | 56.93 ± 0.56 | 68.5 ± 8.9 | 30.24 ± 0.023 |
| | KOH | | 72.41 ± 6.82 (65.60 ± 6.14) * | 5.81 ± 0.56 (0.0) | 14.2 ± 1.33 | 41.72 ± 0.14 | 71.0 ± 6.65 | 21.10 ± 006 |
| | $NH_3$ | | 71.91 ± 6.02 (66.25 ± 6.03) * | 4.66 ± 0.46 (0.0) | 18.5 ± 1.53 | 53.83 ± 0.02 | 92.5 ± 7.65 | 27.04 ± 0.001 |
| | $H_2SO_4$ | | 127.00 ± 11.14 (88.09 ± 9.01) | 35.91 ± 3.01 * (0.0) | 32.3 ± 2.92 | 69.39 ± 0.90 | 161.5 ± 14.60 | 62.23 ± 0.074 |
| | $HNO_3$ | | 126.66 ± 11.23 (85.83 ± 9.45) | 38.83 ± 3.56 * (0.0) | 27.3 ± 2.30 * | 60.83 ± 2.15 | 136.5 ± 11.50 * | 51.88 ± 0.183 |
| | $CH_3COOH$ | 2% | 114.33 ± 10.21 * (92.91 ± 8.45) | 20.42 ± 1.85 (0.0) | 17.5 ± 1.58 | 36.47 ± 0.52 | 87.5 ± 7.90 | 29.11 ± 0.040 |
| | HCl | | 99.20 ± 8.87 * (79.64 ± 7.02) * | 18.56 ± 1.56 (0.0) | 19.5 ± 1.77 * | 47.32 ± 0.06 ± | 97.5 ± 8.85 * | 32.79 ± 0.003 |
| | $H_2O_2$ | | 113.86 ± 10.25 * (74.32 ± 9.21) * | 37.54 ± 2.56 * (0.0) | 21.4 ± 2.17 * | 54.87 ± 0.35 | 107 ± 10.85 * | 43.64 ± 0.024 |
| | NaOH | | 110.00 ± 10.25 * (81.48 ± 7.65) * | 28.52 ± 2.02 * (0.0) | 25.3 ± 2.55 * | 60.76 ± 0.13 | 126.5 ± 12.75 * | 46.67 ± 0.009 |
| | $Ca(OH)_2$ | | 91.26 ± 8.98 * (77.83 ± 7.13) * | 12.43 ± 1.02 (0.0) | 15.6 ± 1.58 | 38.73 ± 0.11 | 78 ± 7.90 | 24.68 ± 0.006 |
| | KOH | | 92.80 ± 7.54 * (79.64 ± 7.02) * | 11.16 ± 1.00 (0.0) | 21.4 ± 1.71 * | 51.30 ± 0.13 | 92.5 ± 8.55 | 33.24 ± 0.007 |
| | $NH_3$ | | 89.60 ± 7.65 * (80.00 ± 7.65) * | 8.60 ± 0.78 (0.0) | 21.4 ± 1.81 * | 51.70 ± 0.14 | 107 ± 9.05 * | 32.35 ± 0.008 |
| | $H_2SO_4$ | | 141.04 ± 12.31 (95.12 ± 9.02) | 40.92 ± 3.45 (0.0) | 43.2 ± 3.82 | 84.44 ± 0.07 | 216 ± 19.10 | 83.16 ± 0.006 |
| | $HNO_3$ | | 133.28 ± 11.24 (88.25 ± 8.12) | 42.03 ± 3.97 (0.0) | 27.8 ± 2.21 * | 59.62 ± 0.13 | 139 ± 11.05 * | 55.49 ± 0.010 |
| | $CH_3COOH$ | 3% | 136.26 ± 11.05 (95.26 ± 9.02) | 39.00 ± 2.98 * (0.0) | 19.5 ± 1.62 * | 39.24 ± 0.04 | 97.5 ± 8.10 * | 37.33 ± 0.003 |
| | HCl | | 121.73 ± 10.23 * (78.01 ± 6.56) * | 40.72 ± 3.87 (0.0) | 24.5 ± 1.92 * | 59.18 ± 0.10 | 122.5 ± 9.60 * | 50.32 ± 0.007 |
| | $H_2O_2$ | | 117.66 ± 10.01 * (60.48 ± 5.13) * | 61.18 ± 4.98 (0.0) | 19.5 ± 1.74 * | 67.56 ± 0.14 | 97.5 ± 8.70 * | 55.52 ± 0.009 |
| | NaOH | | 110.64 ± 5.94 * (50.04 ± 4.89) * | 59.6 ± 5.78 (0.0) | 15.6 ± 0.05 | 59.81 ± 1.34 | 78 ± 0.25 | 46.22 ± 0.091 |
| | $Ca(OH)_2$ | | 94.53 ± 5.62 * (56.17 ± 5.01) * | 37.36 ± 3.28 * (0.0) | 11.7 ± 0.48 | 40.05 ± 0.09 | 58.5 ± 2.40 | 26.44 ± 0.005 |
| | KOH | | 110.86 ± 8.97 * (70.07 ± 6.23) * | 39.79 ± 3.87 * (0.0) | 19.5 ± 1.40 * | 53.69 ± 0.03 | 97.5 ± 7.0 * | 41.57 ± 0.001 |
| | $NH_3$ | | 116.48 ± 10.04 * (49.01 ± 4.02) | 66.47 ± 5.87 (0.0) | 21.4 ± 1.81 * | 83.74 ± 1.20 | 107 ± 9.05 * | 68.94 ± 0.082 |
| | $H_2SO_4$ | | 133.33 ± 11.21 (91.43 ± 8.95) | 41.47 ± 3.67 (0.0) | 38.3 ± 3.14 | 81.59 ± 0.09 | 191.5 ± 15.70 | 75.95 ± 0.007 |
| | $HNO_3$ | | 130.80 ± 11.45 (86.14 ± 8.74) | 44.36 ± 4.01 (0.0) | 27.3 ± 2.35 * | 61.81 ± 0.01 | 136.5 ± 11.75 * | 56.46 ± 0.001 |
| | $CH_3COOH$ | 4% | 124.13 ± 10.97 * (92.40 ± 8.98) | 31.03 ± 2.97 * (0.0) | 19.8 ± 1.70 * | 41.62 ± 0.03 | 99 ± 8.50 * | 36.08 ± 0.002 |
| | HCl | | 116.4 ± 10.05 * (77.14 ± 7.01) * | 38.86 ± 3.28 * (0.0) | 21.7 ± 1.89 * | 54.77 ± 0.14 | 108.5 ± 9.45 * | 44.52 ± 0.010 |
| | $H_2O_2$ | | 119.46 ± 10.92 * (72.40 ± 9.75) * | 46.66 ± 4.03 (0.0) | 17.5 ± 1.66 | 47.04 ± 0.11 | 87.5 ± 8.30 | 39.24 ± 0.008 |
| | NaOH | | 102 ± 10.01 * (52.45 ± 5.01) * | 49.05 ± 4.02 (0.0) | 15.6 ± 1.77 | 57.66 ± 0.17 | 78 ± 8.85 | 41.08 ± 0.011 |
| | $Ca(OH)_2$ | | 87.73 ± 8.01 * (40.28 ± 4.01) | 47.05 ± 4.25 (0.0) | 15.6 ± 1.44 | 75.05 ± 0.10 | 78 ± 7.20 | 45.99 ± 0.005 |
| | KOH | | 105.28 ± 10.02 * (75.2 ± 6.64) * | 30.00 ± 2.76 * (0.0) | 15.6 ± 1.50 | 40.55 ± 0.12 | 78 ± 7.50 | 29.82 ± 0.008 |
| | $NH_3$ | | 117.76 ± 10.98 * (68.57 ± 6.52) * | 49.10 ± 3.98 (0.0) | 19.5 ± 1.99 * | 55.58 ± 0.05 | 97.5 ± 9.95 * | 45.71 ± 0.003 |
| Steam+enzyme treatment | | | 100.64 ± 7.04 (67.2 ± 5.01) | 33.14 ± 2.67 (0.0) | 23.4 ± 1.51 | 67.84 ± 0.22 | 117 ± 7.55 | 47.68 ± 0.009 |

The values in parentheses indicate the glucose yields. All the values differ from the control significantly by the Holm–Sidak test with $p < 0.001$, except those marked with *.

Among the bases investigated, treatment with $Ca(OH)_2$ exhibited the highest release of sugars at a 1% concentration, with a level of $380.35 \pm 30.29$ mg/g. When the concentration was increased to 3%, the amount of released sugars further increased to $472.65 \pm 41.05$ mg/g. However, at a concentration of 4%, there was a gradual decrease in carbohydrate conversion. Similarly, KOH pretreatment followed a similar trend, with $362.05 \pm 31.25$ mg/g of sugars released at a 1% concentration, which improved to $554.30 \pm 42.53$ mg/g at a 3% concentration. These findings are consistent with a previous report on pretreatment techniques using dilute acid ($H_2SO_4$) and dilute alkali (NaOH) for bioethanol production from oil palm empty fruit bunches [42]. The optimal conditions for dilute acid treatment (161.5 °C, 9.44 min, and 1.51% acid loading) resulted in an 85.5% glucose yield. However, the dilute alkali treatment exhibited lower performance under the tested conditions.

Various treatments with different temperatures and concentrations of chemicals, including $H_2SO_4$ (0.5–2.0% *v/v*, 50–121 °C, 1 h), NaOH (0.5–2% *w/v*, 50–121 °C, 1 h), $Ca(OH)_2$ (0.2–4% *w/v*, 50–121 °C, 1 h), and hot water (50–121 °C, 1 h), have been investigated to determine the optimal strategy for cellulose degradation and the production of reducing sugars [30]. The study revealed that the highest cellulose degradation was achieved using a 2% $H_2SO_4$ pretreatment at 121 °C for 1 h, resulting in a significant yield of reducing sugars [43]. It was also observed that the yields of reducing sugars decreased at higher concentrations of acids or bases. This phenomenon could be attributed to increased interactions between amino acids and sugars, leading to Maillard browning reactions at higher temperatures and alkaline conditions, ultimately resulting in the loss of reducing sugars [44]. Additionally, the breakdown of free sugars into furans (such as furfural and hydroxymethyl furfural) and acids (such as levulinic acid and formic acid) at higher temperatures and in high-acid environments could contribute to the lower sugar yields. These products may further degrade, leading to the formation of insoluble carbon-enriched compounds known as chars and/or pseudo-lignin [45–47]. In a study on the pretreatment of wheat straw under various dilute acid conditions, a 1% $H_2SO_4$ concentration yielded the highest glucose yield, reaching 89% of the theoretical maximum [48]. However, contrary to our findings, a study showed that alkaline peroxide pretreatment of wheat straw resulted in the highest sugar concentrations, with 31.82 g/L glucose and 13.75 g/L xylose, outperforming thermal, dilute acid, and dilute base pretreatments [49].

Upon careful analysis of the total reducing sugar and glucose yields in dried potato peels subjected to various pretreatments, it was analyzed that chemical pretreatment using acids or alkalis alone was insufficient for the complete breakdown of the complex carbohydrate structure present in the peels. Thermo-chemical pretreatments combining steam with acids or alkalis were found to be more effective in reducing the crystallinity of various polymers. Among the thermo-chemical pretreatments employed, the use of 3% sulfuric acid in combination with steam was found to be the most effective strategy for the enzymatic hydrolysis of potato peels. This was supported by the release of significant amounts of sugars, with total reducing sugar concentrations reaching $141.04 \pm 12.31$ g/L and glucose concentrations reaching $95.12 \pm 9.02$ g/L, which are higher than any previously reported yields from potato peels (Table 4).

### 3.4. Qualitative Detection of Sugars in the Enzymatic Hydrolysate of Pretreated Potato Peels Using TLC

The enzymatic saccharification of pretreated potato peels resulted in a hydrolysate that was analyzed using thin-layer chromatography. The analysis revealed clear bands corresponding to glucose and xylose, indicating the presence of both C6 and C5 sugars. Traces of mannose were also detected. Figure 1 shows the distinct bands observed in the hydrolysate. Band (a) corresponds to xylose, while band (b) is a mixed band containing glucose, mannose, and arabinose.

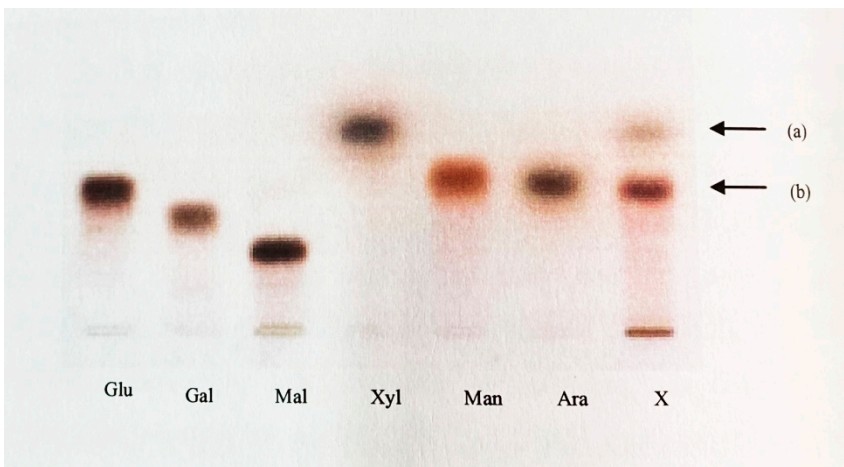

**Figure 1.** Thin-layer chromatogram of enzymatic hydrolysate of potato peels (X) with the standards of glucose, galactose, maltose, xylose, mannose, and arabinose. Band (a) corresponds to xylose, and band (b) corresponds to mixed band of glucose, mannose, and arabinose.

### 3.5. Structural Changes in Untreated, Thermo-Acidic Pretreatment, and Thermo-Acidic Pretreatment Followed by Enzymatically Hydrolyzed Samples of Potato Peels

Scanning electron microscopy (SEM) was used to qualitatively confirm the changes in the morphology of potato peels due to thermo-acidic pretreatment and enzymatic hydrolysis. The untreated potato peel structure appeared solid and intact without visible fragmentation (Figure 2a). However, thermo-acidic pretreatment with enzymatic hydrolysis disrupted the compactness of the structure, resulting in cracks and damage to the biomass (Figure 2b). Furthermore, thermo-acidic pretreatment followed by enzymatic hydrolysis caused the formation of cavities and crinkles, which more effectively damaged the biomass (Figure 2c). These structural changes increased the exposed surfaces and enhanced the availability of cellulose for enzymatic action. This observation is consistent with a previous study by Soltaninejad et al. [7] that demonstrated the disruption of crystallinity and the formation of an amorphous form during enzymatic degradation.

To compare the structural properties of the untreated, thermo-acidic pretreated, and thermo-acidic pretreated + enzymatically hydrolyzed potato peel samples, X-ray diffraction (XRD) analysis was performed in the 0–50θ range using CuKα radiation. Figure 3 shows the XRD pattern with 2θ versus intensity, displaying peaks of cellulose at approximately 2θ = 26. Comparing the peak widths of the samples, it was observed that hydrolysis disrupted the crystalline region of cellulose. The maximum decrease in peak intensity was observed in sample 3 (depicted as red in Figure 3), which underwent thermo-acidic pretreatment followed by enzymatic hydrolysis, indicating a more efficient hydrolysis. Sample 2, which underwent only thermo-acidic pretreatment, showed a narrower peak width and lower intensity reduction compared to sample 3. Sample 1 (untreated potato peels) had the narrowest peak and the lowest intensity reduction (Figure 3). Notably, a significantly wider peak was observed in sample 3 compared to the control and thermo-acidic pretreatment samples, consistent with a similar study by Barampouti et al. [50] on alkaline-pretreated and enzymatically hydrolyzed potato peel waste.

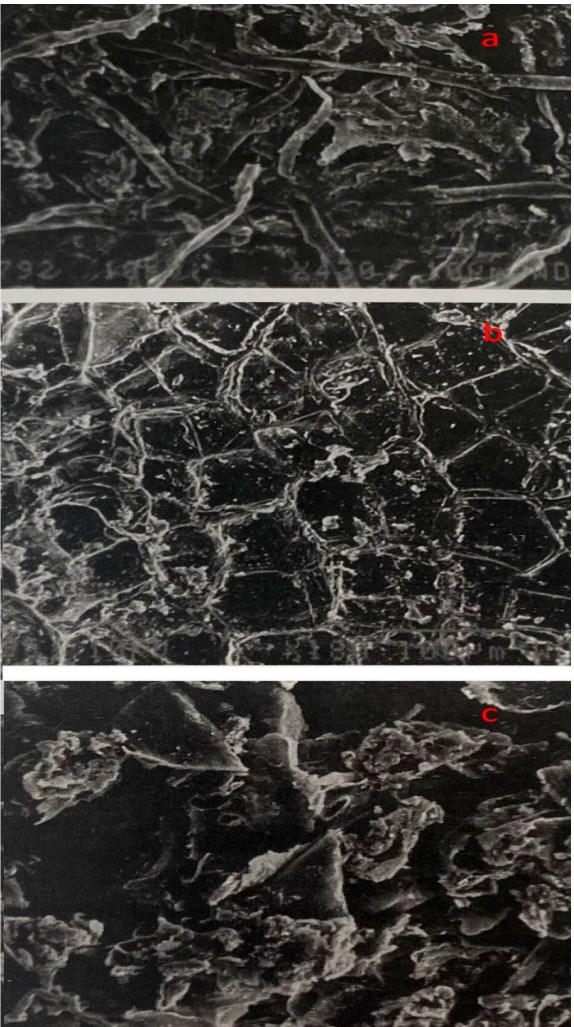

**Figure 2.** SEM micrographs of untreated (**a**), thermo-acidic pretreated (**b**), and thermo-acidic pretreated + enzymatically hydrolyzed (**c**) samples.

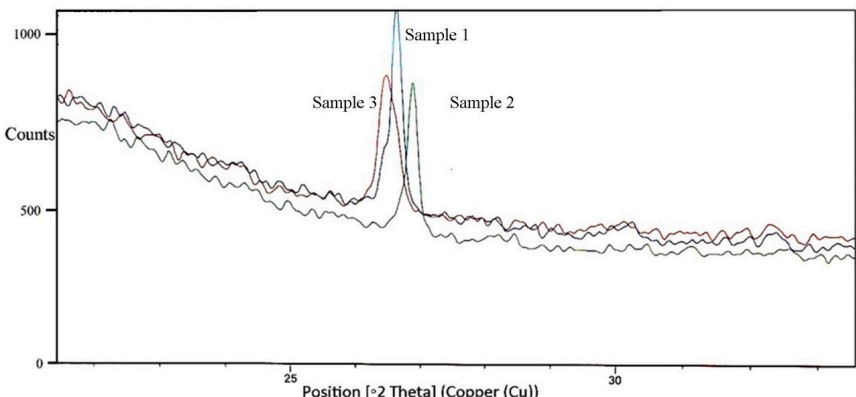

**Figure 3.** XRD pattern of untreated (sample 1), thermo-acidic pretreated (sample 2), thermo-acidic pretreated + enzymatically hydrolyzed (sample 3) samples.

### *3.6. Fermentation of Glucose Released from Potato Peels*

The hexose sugars derived from the enzymatic hydrolysis of potato peels were subjected to fermentation using a distiller's strain of *S. cerevisiae* HT. The highest ethanol yield was achieved from the enzymatic hydrolysate of potato peels pretreated with 3%

$H_2SO_4$ and steam under pressure. The hydrolysate had an ethanol concentration of 43.2 g/L and a maximum ethanol yield of 216 mg/g. However, the hydrolysate obtained from the biomass pretreated with 4% $H_2SO_4$ exhibited a lower fermentation efficiency and ethanol productivity, as shown in Table 4. This decrease in fermentation performance may be attributed to the formation of inhibitory substances during the treatment at high temperatures and acid concentrations. Similar phenomena have been reported in the steam explosion of wheat straw and fermentation of model substrates and hydrolysates by *Pichia stipites* [51]. The severity of $H_2SO_4$ treatment has also been found to impact the sugar yield from spruce (softwood) and the fermentability of the hydrolysate by *S. cerevisiae* [52]. These findings highlight the importance of optimizing the pretreatment conditions to balance the desired sugar release with the inhibitory effects on subsequent fermentation processes. It is crucial to carefully select the pretreatment parameters to maximize the ethanol yield and productivity while minimizing the formation of inhibitory compounds.

The decrease in monosaccharide yield coincided with the highest concentrations of furfural and 5-hydroxymethylfurfural (5-HMF). This suggests that the thermo-chemical pretreatment of potato peels with 3% $H_2SO_4$ resulted in the maximum release of reducing sugars and alcohol during enzymatic hydrolysis and fermentation. This successful validation underscores the potential of potato peels as a significant feedstock for second-generation bioethanol production. Procentese et al. [53] conducted a comprehensive review of various agricultural and agro-food waste residues, including potato peels, to assess their suitability as feedstocks for the production of second-generation biofuels such as ethanol and butanol. They estimated the maximum rates of biofuel production based on feedstock availability, average composition, and reported yields. The study revealed that conventional pretreatment methods could contribute up to 32% of the ethanol and 23% of the butanol of current European biofuel production, whereas innovative pretreatment methods could increase these contributions to 40% of the ethanol and 19% of the butanol.

Several studies have explored the potential of potato peels for biofuel production, including biogas [6,7,54], bioethanol [8,55–59], and xanthan gum [7]. Felekis et al. [8] achieved the highest ethanol concentration of 9 ± 0.9 g/L by pretreating potato peels with 1% *w*/*v* NaOH for 6 h at 50 °C, using commercial enzyme preparations. Achinas et al. [54] investigated the anaerobic digestion of potato peels with and without dilute $H_2SO_4$ pretreatment, observing improved biogas production with pretreatment. Soltaninejad et al. [6] employed the organosolv pretreatment method to produce bioethanol and biogas from potato peel waste (PPW), achieving a maximum bioethanol concentration of 18.04 g/L by pretreating PPW at 180 °C with 75% ethanol and 1% acid. Sivasakthivelan et al. [55] studied the optimal conditions for bioethanol production using *S. cerevisiae*, finding that pH 5.5, temperature of 30 °C, an inoculation of 8%, substrate concentration of 3%, and a maximum time of 48 h resulted in a maximum bioethanol concentration of 11.46 g/L. Atitallah et al. [56] explored various saccharification and fermentation techniques for bioethanol production from potato peel waste, employing thermal and chemical (acid, alkali) pretreatments as well as enzymatic hydrolysis using commercial enzymes (cellulase and amylase) and lab-scale-produced enzymes (α-amylase from *Bacillus* sp. Gb67). The use of commercial enzymes led to a higher saccharification efficiency (72.38%) and ethanol yield (0.49 g/g consumed sugars). Sansui et al. [57] investigated the effect of metallic oxide nanoparticles on ethanol production from potato peel hydrolysate using *S. cerevisiae* BY4743. $Fe_3O_4$ NPs enhanced ethanol production, with a maximum ethanol yield of 0.26 g/g, glucose utilization of 99.95%, and 51% fermentation efficiency. They achieved a maximum ethanol concentration of 5.24 g/L and a maximum production rate of 0.72 g/L/h. Chohan et al. [58] optimized the production of bioethanol from potato peel waste through simultaneous saccharification and fermentation, considering factors such as temperature, pH, and solid loading. Under optimal process conditions of 40 °C, pH 5.78, and 12.25% *w*/*v* solid loading, they observed a maximum bioethanol concentration of 22.54 g/L and a yield of 0.32 g/g. Madadi et al. [59] employed liquid hot water and 5% CaO treatments to enzymatically saccharify potato straw, achieving complete saccharification. Subsequent yeast fermentation

using the saccharified material resulted in a maximum bioethanol yield of 24% based on the dry matter.

Nutrient supplementation has been shown to enhance the growth and metabolism of microorganisms involved in the fermentation process, leading to improved alcohol productivity from potato peels. The impact of nutrient supplementation on the ethanol yield in the present study is presented in Table 5. Among the positive influencers that contributed to enhanced ethanol production, $MgSO_4.7H_2O$, $(NH_4)_2SO_4$, peptone, $(NH4)_2(HPO_4)$, $(NH_4)(H_2PO_4)$, $KH_2PO_4$, $K_2HPO_4$, and $ZnSO_4$ exhibited higher fermentation efficiency, with percentages of 82.63, 84.60, 82.01, 85.86, 87.22, 86.95, 86.81, and 88.38%, respectively, compared to 81.95% for the control. The effect of nutrient supplementation on ethanol production from different feedstocks has also been investigated by various research groups. Suriyachai et al. [60] reported the supplementation of yeast extract (1 g/L), $(NH_4)_2SO_4$ (5 g/L), and $MgSO_4 \cdot 7H_2O$ (0.025 g/L) during ethanol production from rice straw using a co-culture of *S. cerevisiae* and *Scheffersomyces stipitis*, resulting in an ethanol productivity of 28.6 g/L. In a study, the use of ionic liquids was investigated to pretreat pine needle biomass for ethanol production via consolidated bioprocessing (CBP). The process involved the use of the *B. subtilis* G2 enzyme preparation in combination with fermentation by *S. cerevisiae* and *P. stipitis* at a pH of 5.6. The medium was supplemented with yeast extract (1.5 g/L), peptone (1.0 g/L), $(NH_4)_2SO_4$ (1.0 g/L), $K_2HPO_4$ (1.0 g/L), and $MgSO_4$ (1.0 g/L), resulting in an ethanol yield of 0.148 g/g after 72 h of fermentation [61]. Hossain et al. [62] reported the use of Wickerhamia sp. for ethanol production from potato peels. The supplementation of malt extract, tryptone, and $KH_2PO_4$ enhanced ethanol production, resulting in a yield of 21.7 g/L at 30 °C after 96 h of fermentation. In another study, co-cultures of the yeasts *S. cerevisiae* and *P. stipitis* were used to produce ethanol from kitchen waste. The medium was supplemented with $KH_2PO_4$, $MgCl_2.6H_2O$, and $(NH_4)_2SO_4$ at a concentration of 1 g/L each, resulting in ethanol productivity of 45.4 g/L at 30 °C [63]. Zhao et al. [64] utilized corn stalks for ethanol production at 30 °C by employing engineered *S. cerevisiae* strains and supplementing with $MgSO_4$ as a metal ion inducer. They achieved an ethanol yield of 46.87 g/L. Chohan et al. [58] conducted a study on ethanol production from thermally pretreated potato peels by employing *S. cerevisiae* BY4743 and supplementing with yeast extract, peptone, $(NH_4)_2SO_4$, $KH_2PO_4$, and $MgSO_4$, resulting in a maximum ethanol productivity of 22.54 g/L.

In the present study, it was observed that $ZnSO_4$ at a concentration of 0.1% *w/v*, along with a 20% solid loading, yielded the most effective results, with a production of 48.1 ± 3.88 g/L of ethanol and a fermentation efficiency of 88.38 ± 2.49%. This corresponded to a yield of 24.05% based on dry matter. $(NH_4)(H_2PO_4)$, also at the same concentration, showed favorable outcomes, with a yield of 46.8 ± 3.84 g/L of ethanol. Peptone supplementation resulted in a yield of 44.0 g/L of ethanol. Additionally, different solid loading levels (22% and 24% *w/v*) were tested in combination with the supplementation of peptone, $(NH_4)(H_2PO_4)$ and ZnSO4 at 0.1% *w/v* each. These combinations yielded significantly good results, with 51.67 ± 4.35 g/L (23.4% based on dry matter and 88.7 ± 1.84% fermentation efficiency) and 54.75 ± 4.45 g/L (22.81% based on dry matter and 87.11 ± 1.23% fermentation efficiency) of ethanol, respectively. The highest yield and efficiency were observed with a 22% substrate loading, although there was a slight decrease in the case of the 24% loading. However, the ethanol concentration was higher in the latter (Figure 4).

This study demonstrated that the production of ethanol from potato peel waste using in-house-produced enzymes resulted in a lower production cost of USD 0.65 per liter, compared to the cost of USD 1.50 per liter when using commercial enzymes from Advanced Enzymes (India). Additionally, it was observed that a substantial amount of C5 residual sugars remained after fermentation with *S. cerevisiae*, indicating the potential for further improvement in the yield by employing a suitable consortium of yeasts capable of fermenting both hexose and pentose sugars.

**Table 5.** Ethanol yields as a result of supplementation of various nutrients with varying solid loadings during fermentation.

| Nutrient | Total Reducing Sugars (% g/L) | Total Glucose (% g/L) | Residual Reducing Sugar (% g/L) | Residual Glucose (% g/L) | Ethanol (g/L) | Fermentation Efficiency (%) |
|---|---|---|---|---|---|---|
| 20% solids | | | | | | |
| Control | 141.0 ± 10.05 | 93.5 ± 8.45 | 38.3 ± 1.45 | 0 | 43.0 ± 3.42 | 81.95 ± 4.13 |
| $MgSO_4.7H_2O$ | 141.0 ± 10.05 | 93.5 ± 8.45 | 37.5 ± 1.66 * | 0 | 43.7 ± 3.47 * | 82.63 ± 1.69 * |
| $(NH_4)_2SO_4$ | 141.0 ± 10.05 | 93.5 ± 8.45 | 36.9 ± 1.55 * | 0 | 45.0 ± 3.60 * | 84.60 ± 1.72 * |
| Yeast Extract | 141.0 ± 10.05 | 93.5 ± 8.45 | 38.4 ± 1.21 * | 0 | 40.0 ± 3.38 * | 76.29 ± 1.47 |
| Urea | 141.0 ± 10.05 | 93.5 ± 8.45 | 39.0 ± 1.35 * | 0 | 41.0 ± 3.55 * | 78.66 ± 1.19 * |
| Peptone | 141.0 ± 10.05 | 93.5 ± 8.45 | 36.0 ± 1.77 * | 0 | 44.0 ± 3.42 * | 82.01 ± 1.18 * |
| $(NH_4)_2(HPO_4)$ | 141.0 ± 10.05 | 93.5 ± 8.45 | 35.7 ± 1.29 * | 0 | 46.2 ± 3.76 * | 85.86 ± 1.86 * |
| $(NH_4)(H_2PO_4)$ | 141.0 ± 10.05 | 93.5 ± 8.45 | 36.0 ± 1.45 * | 0 | 46.8 ± 3.84 * | 87.22 ± 0.16 |
| $KH_2PO_4$ | 141.0 ± 10.05 | 93.5 ± 8.45 | 35.0 ± 1.20 * | 0 | 47.1 ± 3.84 * | 86.95 ± 2.04 |
| $K_2HPO_4$ | 141.0 ± 10.05 | 93.5 ± 8.45 | 34.6 ± 1.13 | 0 | 47.2 ± 3.87 * | 86.81 ± 1.91 |
| $ZnSO_4$ | 141.0 ± 10.05 | 93.5 ± 8.45 | 34.5 ± 1.21 | 0 | 48.1 ± 3.88 * | 88.38 ± 2.49 |
| 22% solids | | | | | | |
| Peptone+ $(NH_4)(H_2PO_4)$+ $ZnSO_4$ | 155.0 ± 11.45 | 101.6 ± 9.21 | 41.0 ± 1.65 * | 0 | 51.67 ± 4.35 * | 88.70 ± 1.84 |
| 24% solids | | | | | | |
| Peptone+ $(NH_4)(H_2PO_4)$+ $ZnSO_4$ | 169.0 ± 11.70 | 108.1 ± 9.45 | 46.0 ± 1.56 | 0 | 54.75 ± 4.45 | 87.11 ± 1.23 |

All the values differ from the control significantly by the Holm–Sidak test with $p < 0.001$, except those marked with *.

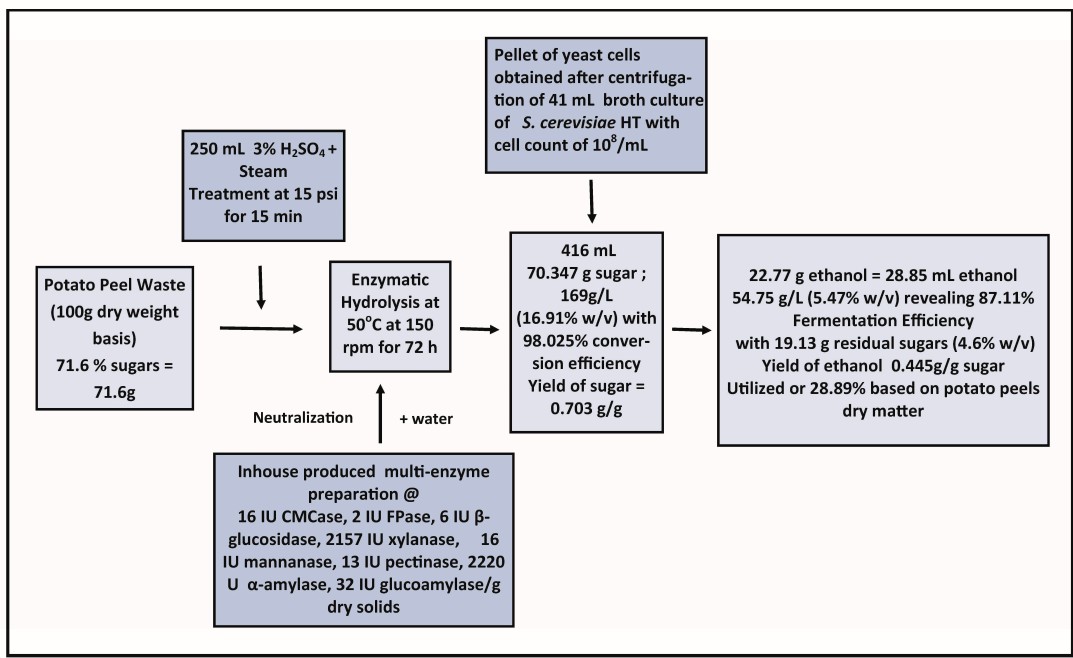

**Figure 4.** Mass balance after thermo-acidic treatment followed by enzymatic hydrolysis and fermentation of potato peel waste for ethanol production.

## 4. Conclusions

This study successfully demonstrates the effectiveness of an internally developed enzyme cocktail in efficiently hydrolyzing acid-pretreated potato peels, followed by fermentation using a distiller's strain of *S. cerevisiae* for bioethanol production. The study achieved a remarkable ethanol productivity of 54.75 g/L by supplementation with peptone + $(NH_4)(H_2PO_4)$ + $ZnSO_4$, surpassing previous investigations that either used low substrate loading, higher pretreatment temperatures, commercial enzyme preparations, or obtained low ethanol concentrations, resulting in increased process costs. These results provide a promising basis for further scalability studies and the commercialization of this technology for bioethanol production using an affordable biorenewable resource in the form of potato peels. Moreover, the reported yields were achieved using a hexose-fermenting yeast strain, suggesting potential improvements in alcohol yield by employing an appropriate consortium of both hexose- and pentose-fermenting yeast strains to ferment the mixture of hexoses and pentoses generated through enzymatic hydrolysis.

**Author Contributions:** S.K.S.: conceptualization, methodology, writing—review and editing, supervision; B.S.: investigation, methodology, writing—original draft preparation, writing—review and editing; A.S.: investigation, methodology, writing—original draft preparation, writing—review and editing; B.T.: investigation, methodology, writing—original draft preparation, writing—review and editing; R.S.: investigation, methodology, writing—original draft preparation, writing—review and editing, supervision. All authors have read and agreed to the published version of the manuscript.

**Funding:** This research was funded by Panjab University, Chandigarh, India.

**Institutional Review Board Statement:** Not applicable.

**Informed Consent Statement:** Not applicable.

**Data Availability Statement:** Not applicable.

**Conflicts of Interest:** The authors declare no conflict of interest.

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
