# Peer review of "Exploring the Potential of Potato Peels for Bioethanol Production through Various Pretreatment Strategies and an In-House-Produced Multi-Enzyme System"

_sustainability, doi:10.3390/su15119137_

Round 1

Reviewer 1 Report

This study evaluated the potential of potato peels for bioethanol production, using different pretreatment methods and in-house produced multi-enzyme. 3% H2SO4 followed by steam pretreatments were the most effective pretreatment methods that produced 141 g/L sugar and 43.2 g/L ethanol at 20% biomass loading. Some point needs to be considered before publication.

1.     The title can be modified to “Potential of potato peels to bioethanol using different pretreatments strategies and in-house produced multi-enzyme system.

2.     Change the term “bio ethanol” to “bioethanol”, in the whole paper.

3.     Change the term “pre-treatment” to “pretreatment”

4.     Keywords need to be from the words in the title.

5.     The following references can be used to improve the introduction part: for example for lines 40-56 (https://doi.org/10.1016/j.carbpol.2021.118070),

 (https://doi.org/10.1016/j.indcrop.2023.116815).

6.     What about the biomass loading for the pretreatment step? How much percentage/

7.     Which method did you used to detect the chemical composition of potato peels?

The English language can be improved 

Author Response

We sincerely appreciate your diligent evaluation and constructive criticism, which have undoubtedly helped us improve the overall quality of the manuscript.

Reviewer 2 Report

The submitted article for review entitled "Potato peels to bioethanol: Evaluation of different pre-treatment strategies for low-cost saccharification using in-house produced multi-enzyme system" is written correctly and meets the requirements of scientific articles. The authors of the paper examined a wide range of possibilities for bioethanol production from potato peels. I have minor remarks: Fig. 1 is of poor quality In table 5, the significance of differences is marked with *, however, there are no markings in the table, Fig. 4 Please fit the drawing into the text Conclusions should be supplemented with information on how much bioethanol production has increased compared to the traditional method and how fermentation efficiency has increased

Author Response

(The authors gave the same response as above.)

Reviewer 3 Report

General remarks:

This manuscript aims to discuss the conversion of waste potato peels into bioethanol through several pretreatment approaches followed by the fermentation using in-house produced multi-enzyme. Overall, the study showed some contribution to the literature with current findings and discussion that includes several pretreatment methods, enzyme hydrolysis and the final mass balance study. However, the manuscript is not well-presented and further improvement on the structure, methodology and result discussion should be carried out for a better flow of the manuscript. More revision will be needed to improve and restructure the manuscript for better understanding.

Specific remarks:

1.         Please elaborate more on the significance and functionality of the in-house enzymes compared to commercially available enzymes.

2.         Section 2.3.1: How the pressure and time of thermal pretreatment conditions are selected?

3.         Please elaborate on how the potato peels are turned into powder form and any specific particle size.

4.         Section 2.4: Why the steam-pretreated sample was selected as the control for enzymatic hydrolysis?

5.         Any reference method for TLC sugar analysis?

6.         Please provide further details on the SEM and XRD analysis.

7.         Section 3.2: Any supporting data on the differences between lignin and hemicellulose content before and after pretreatment?

8.         Please elaborate on the calculation of carbohydrate conversion efficiency.

9.         Please standardize the unit of reducing sugar content in g/L.

10.       Please clarify the conversion efficiency in Tables 1-4.

11.       Perhaps can convert the data in Tables 1-4 into Figures for better presentation and understanding.

12.       Kindly double check the spelling, grammar, typing errors and formatting of the manuscript.

13.       How the ethanol yield is determined and calculated?

14.       Please improve the presentation of Table 3&4.

15.       Please revise the captions of Fig 2 & 3.

16.       Please improve Figure 4 and the mass balance calculation.

The manuscript shall be double-checked and revised in terms of language.

Author Response

(The authors gave the same response as above.)

Round 2

Reviewer 1 Report

The authors modified the manuscript based on the reviewers comments, and it can be accepted in its current format. 

Author Response

We appreciate your insightful comments and suggestions, which have helped us improve the quality of our paper.

Reviewer 3 Report

Overall, the revision has been made based on the comments appropriately and more details were given to improve the manuscript.

Please find some minor comments for the manuscript:

1. Kindly make sure all the scientific names are in italic form.

2. Line 152: The concentration of total reducing sugar is expressed in g or g/L? 

3. In Line 199 and 208, are both fermentation efficiency in %? What is the difference?

4. Please take note of the spacing between °C.

Author Response

(The authors gave the same response as above.)
